# Beyond Biodiversity: Can Environmental DNA (eDNA) Cut It as a Population Genetics Tool?

**DOI:** 10.3390/genes10030192

**Published:** 2019-03-01

**Authors:** Clare I.M. Adams, Michael Knapp, Neil J. Gemmell, Gert-Jan Jeunen, Michael Bunce, Miles D. Lamare, Helen R. Taylor

**Affiliations:** 1Department of Anatomy, University of Otago, 270 Great King Street, Dunedin, Otago 9016, New Zealand; michael.knapp@otago.ac.nz (M.K.); neil.gemmell@otago.ac.nz (N.J.G.); gjeunen@gmail.com (G.-J.J.); helen.taylor@otago.ac.nz (H.R.T.); 2Trace and Environmental DNA (TrEnD) Laboratory, School of Molecular and Life Sciences, Curtin University, Bentley, Perth, WA 6102, Australia; michael.bunce@curtin.edu.au; 3Department of Marine Science, University of Otago, 310 Castle Street, Dunedin, Otago 9016, New Zealand; miles.lamare@otago.ac.nz

**Keywords:** sampling methodology, mtDNA, mitochondrial DNA, conservation, biodiversity, populations, genetics, eDNA

## Abstract

Population genetic data underpin many studies of behavioral, ecological, and evolutionary processes in wild populations and contribute to effective conservation management. However, collecting genetic samples can be challenging when working with endangered, invasive, or cryptic species. Environmental DNA (eDNA) offers a way to sample genetic material non-invasively without requiring visual observation. While eDNA has been trialed extensively as a biodiversity and biosecurity monitoring tool with a strong taxonomic focus, it has yet to be fully explored as a means for obtaining population genetic information. Here, we review current research that employs eDNA approaches for the study of populations. We outline challenges facing eDNA-based population genetic methodologies, and suggest avenues of research for future developments. We advocate that with further optimizations, this emergent field holds great potential as part of the population genetics toolkit.

## 1. Introduction

The study of population genetics describes changes in the genetic diversity of populations through evolutionary mechanisms such as natural selection, genetic drift, mutation, and gene flow, the totality of which have evolutionary consequences and can lead to speciation (e.g., [1,2,3,4,5,6]). The direction, magnitude and mechanisms of population connectivity can be examined by quantifying gene flow (e.g., [4,5,6]). Furthermore, genetic data can be used to estimate inbreeding, which has potential fitness consequences (e.g., [7,8,9]). Understanding the genetics of populations can help us understand the past distributions [10], present status [11] and future prospects of species [12]. Genetic data are, therefore, useful for informing conservation management strategies.

Population genetic research requires genetic sampling from organisms of interest. For example, blood and tissue samples have traditionally been sources of genetic material for large marine mammals, e.g., flipper clips from seal populations [13]. However, sampling tissue has negative effects on the target organism, such as: lethal sampling [14], causing harm or discomfort [15], disfigurement [16], or imposing stress [17]. Sampling can also be expensive, dangerous and time consuming for researchers [18,19,20]. For example, studying venomous reptiles may put researchers at risk of a bite [21] and traditional sampling efforts to directly encounter the target species may be more costly than reagents required for eDNA analysis [22]. Furthermore, great sampling effort is required, especially if the target species is rare or difficult to find; in one instance taking >90 person-hours to find a single silver carp (*Hypophthalmichthys molitrix*) [19]. Many statistical models have been conceived to understand rare species ecology based on limited sample sizes [23,24,25]. More recently, non-invasive sampling techniques have been developed to reduce harm and expense [26]. In marine environments, for instance, fecal plumes, respiratory blow, parasites, and shed skin have proven to be useful DNA sources [27,28,29,30,31]. Such indirect samples have been used to amplify mitochondrial DNA (mtDNA) and nuclear DNA (nuDNA) markers, allowing measurement of allelic diversity, kinship, sex ratios, abundance and effective population size, density, evolutionary significant units, mating systems, and patterns of dispersal in species that might otherwise be difficult to sample [29,30,32,33,34,35,36,37,38,39]. The advent of promising new non-invasive technologies, such as environmental DNA (eDNA), creates further options for non-invasive population genetic sampling.

Here we adopt the definition of eDNA from Taberlet et al. [40] defining it as a “*complex mixture of genomic DNA from many different organisms found in an environmental sample.*” However, we take a narrower view of environmental sample; defining eDNA as DNA taken from soil, water, or air, excluding samples from direct individual remains such as fecal matter, hair, or feeding traces [18,37,41]. Direct organismal traces may allow for assignment of genetic data to specific individuals [41,42,43]. From these non-invasive samples, DNA may be used to describe abundance [35,36,37], effective population size [44], density [38], diet [45], and sex of individuals [46,47]. Additionally, estimates of genetic diversity (e.g., inbreeding within a population) [48,49], gene flow [50,51], evolutionary significant units [39] and mating systems (e.g., multiple paternity) [32] have been obtained. However, eDNA samples from soil, water, or air are useful when individual traces cannot easily be identified and sampled. For example, eDNA can be used to target sites of suspected occupancy before intensive, invasive sampling effort is carried out in difficult-to-sample habitat [52]. With eDNA metabarcoding of environmental samples, the use of traditional methodology in tandem with eDNA methodology may reveal more biodiversity than either method alone [53].

Environmental DNA is already being used to improve biodiversity estimates (e.g., species-level taxa), when a complete snapshot of biodiversity is difficult to obtain via traditional methods (e.g., transects, video, trawl nets, trapping) [54,55,56,57]. The sensitivity of eDNA methods makes them ideal for detecting the presence of endangered, low-density invasive, transient, and cryptic species [58,59,60,61,62,63], especially when sampling efforts to detect low density species would be unmanageable [19]. Environmental DNA methods are not without challenges, but because eDNA offers sampling advantages such as sensitivity, simplicity and reduced harm, it is already being employed across a variety of biomes as a biodiversity monitoring tool, frequently in tandem with traditional methodologies [64,65,66,67]. In addition to biodiversity monitoring, eDNA could potentially be used for population genetic studies, mitigating some of the issues around invasive sampling. Population genetic eDNA is a newly emerging field that offers exciting prospects, but at present comes with challenges, discussed below [42,68,69,70,71,72].

### Why Use eDNA over Direct Sampling for Population Genetics?

Some of the species in eDNA population genetic studies are of conservation and management interest and can be difficult to sample [68,69,71]. Sampling protected species (e.g., endangered or threatened species) and species of cultural significance requires permitting processes that cost time and effort [73,74,75]. Data deficient species may not have a good sampling framework in place [76]. Species such as cetaceans may travel widely, while stranded individuals (often a major source of tissue samples) may not correspond to a population’s usual distribution [69]. Although studies using eDNA methods for population genetics to date have been heavily focused on the marine biome, eDNA could also be useful for other difficult-to-sample species in various habitats. For example, eDNA has been used to detect the presence of endangered species such as the great crested newt (*Triturus cristatus*) in freshwater habitat [58,77,78]. Terrestrial endangered species, such as the Bornean orangutan (*Pongo pygmaeus*) and Sunda pangolin (*Manis javanica*), may leave eDNA traces in locations such as salt licks [79]. Given the elusiveness of some imperiled organisms, eDNA may be one of the best options for describing genetic variation in specific cases. Understanding the population dynamics of these species is important for effective conservation management, especially if they fall under a protected status [80].

Not only can eDNA be useful for managing imperiled species, but invasive species may also be suitable targets. For instance, freshwater invasive species such as bullfrogs (*Lithobates catsebeianus*) and Asian carp (*Hypophthalmichthys* sp.), among many others, have already been successfully detected with eDNA methods [19,81,82,83]. In addition to presence, population genetic eDNA methods may be useful for identifying source populations and pathways of invasion spread, as previous population genetic studies have done [19,84]. We anticipate that developing a widely usable population genetic eDNA toolkit could facilitate monitoring for these invasive species in the future. By sensitively detecting an invasion front and source population, managers may further be able to prevent damage and associated economic costs [85,86]. However, significant sequencing, bioinformatic, and statistical challenges will need to be overcome to go beyond proof-of-concept studies to answer questions about gene flow, genetic drift, mutation, and natural selection. These challenges, discussed below, include but are not limited to: distinguishing individuals, allelic dropout and false alleles related to amplification error, difficulty in relating eDNA to biologically relevant abundances, and expanding to nuclear markers.

## 2. Environmental DNA for Population Genetics

The use of eDNA in population genetics is in its infancy, but a handful of notable studies demonstrate the potential of eDNA to obtain within-species population genetic data. A recent landmark study used eDNA from seawater to examine the mtDNA haplotype variation of whale sharks (*Rhincodon typus*), to assess shark population structure, and their trophic interactions [68]. This work demonstrated that eDNA sampling in areas of known whale shark presence could detect some haplotypes that were also known from directly sampled tissues. Data from eDNA suggested a recently identified Qatar population of whale sharks was more closely related to the Indo-Pacific whale shark aggregation compared to the Atlantic aggregation [68]. In addition to haplotypic variation, this study reported a positive eDNA copy number correlation between quantities of whale shark eDNA and quantities of eDNA from their mackerel tuna prey (*Euthynnus affinis*) over two years, likely reflective of food web interactions [68]. This research suggests avenues for further eDNA mtDNA haplotype studies, especially that of examining population-specific haplotypic variation.

More recently, seawater eDNA work on Northeast Pacific killer whales (*Orcinus orca*) explored whether eDNA-obtained haplotypes could be assigned to known haplotypic variation, and how long genetic material can be detected after known target animal presence [71]. Using droplet digital PCR (ddPCR), a sensitive PCR technique for the absolute quantification of DNA copy number in a sample, killer whale eDNA was detected up to two hours after individuals were observed in the sampled area [71]. Sequencing of these eDNA samples also identified a haplotype belonging to the southern resident killer whale ecotype, a genetically and usually geographically distinct variation of this species [71], showing that eDNA can be used to detect the genetic diversity of a vagile species like killer whales even after they have passed through an area. It may therefore be possible to detect ecotype movements and interactions through eDNA, genetically confirming visual identification.

Beyond assignment to known haplotypes, eDNA has been used to uncover previously unknown intraspecific genetic diversity. For example, previously unknown harbor porpoise (*Phocoena phocoena*) mtDNA control region haplotypes have been detected via seawater eDNA [69]. These haplotypes differed from established haplotypes by one base pair, a small difference, and added additional support to the overall population structure previously described among the different sampling locations for this species [69]. The population structure as resolved with the assistance of eDNA derived haplotypes, supports managing the southeast Alaskan harbor porpoise as two separate populations [69]. Importantly, this study featured positive controls (four known harbor porpoise Sanger-sequenced samples) that helped lay the groundwork for strict quality filtering of the sequence data [69]. The inclusion of these sequences allowed for baseline PCR and sequencing error rates to be detected. This strict quality filtering has been emphasized in other studies [63,68,69,70,72].

In addition to single-species targets, eDNA metabarcoding primer sets have been used to identify multiple haplotypes across multiple species simultaneously [70]. For instance, multiple arthropod mtDNA haplotype ratios have been identified using eDNA from freshwater streams [70]. After strict data filtering and only considering haplotypes found in multiple replicates, between 177 to 200 operational taxonomic units (OTUs), were recovered with the average number of detected haplotypes for each OTU ranging between 2.40 and 3.30 [70].

Separately, 16S rDNA eDNA metabarcoding primers described multiple intraspecific haplotypes from the *Lethrinus* (snapper) genus from seawater samples, also using strict sequence read filtering [72]. Both studies required primers that gave enough resolution to discern intraspecific haplotypes while also being broad enough to capture multiple species [70,72]. These studies help build a framework for determining multi-taxa haplotypic diversity with eDNA metabarcoding technology, but more investigation is needed for other environments and taxa [70,72]. Importantly, the error profiles around each amplicon need to be carefully considered to differentiate rare haplotypes from sequencing error [87].

## 3. Challenges Facing Environmental DNA Population Genetics

Despite recent success and future potential, eDNA population genetic methods require fine tuning before they can be widely applied. Perhaps one of the biggest obstacles in transitioning from species detection to population genetics using eDNA is assigning sequences to individuals. Previous non-invasive genetic techniques for terrestrial organisms have been able to assume different individuals from separate scats [37,48] or feeding sites [42,43,88]. Additionally, DNA extracted from invertebrate stomachs (iDNA) has also been used to obtain host DNA [31,89,90,91]. Indeed, whale shark population structure was confirmed using iDNA from a whale shark copepod (*Pandarus rhincodonicus*) parasite [31]. However, capture of whole, discrete fecal matter beyond initial deposition, feeding traces, or parasites may be challenging in environmental sampling scenarios. Currently, environmental samples cannot parse apart individuals. The same haplotype from a sample could indicate one individual or multiple individuals with the same haplotype. However obtained, confidence in individually-sorted genetic data is important for downstream analysis (e.g., *adegenet* for R, or Bayesian Analysis of Population Structure (BAPS) [92,93]) as population genetic theory concerns the existence and change of genetic variation within individuals between and among populations.

Even when sequences are obtained in sufficient quantities and can be assigned to individuals, many non-invasive genetic sampling techniques, including eDNA, suffer from amplification challenges. Allelic dropout (the loss of allelic variation during sequencing), and false alleles (the apparent presence of non-existent alleles in samples), are notable examples [41,49,94]. Polymerase errors may inflate marker variation, leading to false alleles [95]. As in any population genetic study, regardless of DNA source, missing or false alleles will bias genetic diversity estimates with implications for further analyses [96]. Degraded DNA samples yielding low quality and/or template can contribute to these challenges, requiring increased replicates, increased PCR cycles, or increased sequencing depth [97,98,99]. Recent eDNA sampling has shown that allelic variation can be missed in multiple markers depending on extraction and capture methodologies [43]. Therefore, testing methodologies and using multiple markers will be important for reliable, reproducible eDNA capture and amplification in eDNA population genetic studies.

Some techniques for mitigating quantity challenges for eDNA samples, such as PCR, are not without biases which may impact allelic abundance [98,100,101]. For example, PCR may exponentially amplify common DNA, decreasing the signals of rare genetic variation [102]. When using next-generation sequencing (NGS), there is a fine line between NGS error rates and low-quantity alleles when filtering and analyzing data [70]. The inclusion of positive controls and stringent sequence read filtering could act as a guideline to determine real genetic variation [69,103]. Multiple sampling replicates could be used to further verify the legitimacy of low-abundance genetic variation [69,72]. The same sequencing errors likely will not occur over multiple sampling replicates, and these reads could be considered, even if present at low copy number. Additionally, read depth or eDNA copy number may not to correlate with known biomass [98,104,105]. However, a more recent study indicates strong evidence that flow-corrected eDNA abundance reflects salmon count abundance in two species across two years and life stages [106]. Relating allelic abundance to eDNA genetic variation is necessary to understand allelic frequency differences between and within populations. Correlation correction factors may help, but for multicellular species obtained from environmental sampling, variation in individual rates of DNA shedding may obscure true haplotype ratios [107,108,109].

Furthermore, the number of reads from eDNA-obtained genetic data for target species must be sufficient enough to be distinguishable from background noise to ensure confidence that true genetic variation has been obtained. If initial DNA copy numbers are insufficient (e.g., <200 copies/L, [61,110]), PCR may be too stochastic to amplify genetic variation reliably, or may not amplify it at all. This issue of low copy number (LCN) has been addressed previously in the forensic and ancient DNA (aDNA) fields with technological advancements and modified protocols (e.g., qPCR, [111]; increased reagents, [112]) and awareness of LCN difficulties [113]. Sufficient sampling may increase chances of capturing target eDNA, and limits of detection have been explored for eDNA at the species level [114,115,116]. Ensuring confidence in genetic assignments with sufficient copy number is necessary to have confidence in data.

Increasing the length of sequences and diversity of marker types that can be obtained from eDNA would also be advantageous for answering population genetic questions. Thus far, eDNA research has focused primarily on short (100–400 bp) mtDNA fragments. Additionally, commonly barcoded regions (e.g., *cytochrome oxidase I*) in mtDNA are widely available in public databases for many metazoans [117,118]. One reason for choosing mtDNA is its high copy number availability in environmental samples [119,120,121]. Further, as eDNA degrades over time and space, short (100–400 bp) mtDNA sequences are more likely to be available in environmental samples [122,123,124,125], although larger (>16 kb) mtDNA fragments have recently been obtained from eDNA [126]. However, it would be useful to expand into other, longer molecular markers as the exclusive use of short, maternally-inherited mtDNA fails to incorporate non-maternal genetic variation for most organisms [127,128]. For a deeper understanding of population structure, nuDNA data should be considered [129,130]. Phylogenetically, nuDNA may show different patterns of divergence within populations compared to mtDNA data [127,130,131]. For instance, nuclear variation may show evidence of introgression when mtDNA does not, which could be important to understanding how species and populations are interacting [132]. Additionally, nuclear copies of mitochondrial DNA (*numts*) occur in eukaryotes and can influence population genetic inferences [133,134]. *Numts* are difficult to recognize and have been shown to inflate diversity estimates when used with barcoding techniques [135,136,137]. Obtaining more markers is easier with nuDNA (e.g., 10′s of microsatellites or 10,000′s of SNPs) and can also help provide the ability to distinguish individuals with similar genetic signatures [43,138,139,140]. Multi-allelic microsatellites could be used, and they have the added advantage of establishing a minimum number of individuals present in an eDNA sample. For example, in diploid species, if three alleles are found in a mixture, at least two individuals must exist in that environment [141,142]. Furthermore, markers that identify sex in genetically sex-determined species could give insight into the presence of sexes present in non-sexually dimorphic species.

While much research is focused on obtaining, interpreting, and quantifying these data in an environmental context, there is a clear need for a robust statistical framework for answering population genetic questions from environmentally obtained samples [61,143,144]. Most population genetic theory, and thus, statistical software and tools, is founded upon the expectations and observations of individual diversity [92,93]. The inability to readily distinguish between individuals using eDNA is a current limitation [145]. To complicate the matter, environmental samples may have different, unknown amounts of allelic variation from multiple individuals from different species, which are unlikely to be in equimolar concentrations [107,145]. Unequal concentrations of allelic diversity would be difficult to correct for, given environmental stochasticity and our current understanding of the physicality of eDNA (which is that environments vary and eDNA accumulation, transport, and degradation vary along with them) [146]. It is unlikely that individual heterozygosity will be able to be obtained with eDNA, as eDNA often contains fragmented DNA that cannot be assigned to a specific cell. If individual cells cannot be obtained, population genetic questions using environmental samples will be limited to primarily comparing allelic variation at between different populations [69,145].

Due to the aforementioned challenges, and the emergent nature of the eDNA field, population genetic-oriented eDNA sampling methodologies currently lag behind those of contemporary population genetic methods. Population genetic studies now regularly use thousands of single nucleotide polymorphisms (SNPs) for multiple individuals, not only for comparison of populations across space and year-to-year variation, but also to infer evolutionary histories with coalescence models [147]. Additionally, whole genome sequencing (WGS) is increasingly used to examine population-level genetic variation because it offers very high genomic resolution for detecting selection and the genetic basis of phenotypes (Figure 1) [148]. Already, population-level data collected via RADseq and other reduced-representation sequencing approaches can compare exomes or SNP-based genotypes to answer population-level questions (Figure 1) [149,150,151,152]. It is likely that eDNA population genetics will continue to lag behind traditional population genetics while the aforementioned challenges are being addressed. However, exciting technological developments suggest that at least some of these challenges can be overcome.

## 4. Further Developments and Future Technologies

For all eDNA population genetic studies, a robust sampling design will help minimize LCN DNA profiles. Choosing the correct substrate for sampling a target organism based on ecological knowledge, such as the position in the water column with the highest concentration of DNA of the target organism, will maximize potential copy number [153]. On this front, collecting eDNA may become easier with automated sampling systems for standardized, long-term sampling. After this, carefully considering the extraction methodology, assay, sequencing, and bioinformatic approaches best for the biological question asked is key, as method choice influences genotypes obtained [98,154]. We suggest avenues of further research to move the field forward in Table 1.

Several technologies that could help address the difficulty of assigning genetic information from eDNA samples to individuals, are already available or in development. One way forward might include single-cell sequencing (Table 1). As eDNA contains sloughed cells, collecting and sequencing whole, individual cells could facilitate extraction of individual-level genetic variation [146]. Cells could be collected using flow cytometry and microfluidic technology [155]. From there, extracted DNA from these cells can be sequenced using a variety of sequencing tools (e.g., Illumina, Nanopore) [156]. Advances in droplet microfluidics could potentially facilitate the capture and sequencing of DNA from single cells in parallel [157,158]. For instance, microbial single cell sequencing has already been used to characterize the genomes of marine bacterioplankton after an oil spill [159]. In general, single-cell sequencing technology remains very much the domain of medical and model research systems and there are still issues with biases [160,161]. Small quantities of DNA can result in allelic dropout and low sequencing coverage [160,162]. However, bioinformatics tools to address single-cell sequencing challenges are growing and could be developed for eDNA techniques to facilitate the use of single-cell sequencing for obtaining individual variation from eDNA samples.

Good sampling design, in tandem with technological developments, could also help mitigate issues around obtaining allelic ratios and abundance [154,163]. Increasing sequencing depth also helps to guard against allelic dropout [164]. Previous work has also shown increased PCR replicates and increased sequencing depth increases the number of species detected in samples (Table 1) [99]. However, increasing sequencing may also increase noise and incongruency between PCR replicates [98]. Additionally, the inclusion of positive controls during sequencing or PCRs can assist data processing, serving as a benchmark for separating true haplotypic diversity from sequencing noise where NGS is used [69]. This will aid in determining appropriate filter parameters to ensure strict filtering for high quality data without being too restrictive or ignoring real biological signals. Droplet digital PCR (ddPCR) could also be used to address allelic and copy number abundance in a sensitive way (Table 1) [165,166]. This technology is already being explored for abundance estimations of species using eDNA techniques and could potentially quantify allelic diversity [61,71,165,166]. Other ways to get at quantification of allelic copy number include a correction factor depending on cell type, size, life-history stage and species, which could be integrated into single-cell methodology [107]. Lastly, the more that is understood about the ecology of eDNA in a field setting, the better we can understand and model eDNA abundance [106,146].

Another challenge for population genetic eDNA methods will be to fully develop nuDNA markers for the species of interest using eDNA sampling techniques. Capture methodology, where bait molecules bind to target DNA of interest, is currently being explored for eDNA with some success [42,167]. This methodology can target specific SNPs in nuDNA and may “fish out” extracellular, degraded eDNA (Table 1) [167]. Beyond current technologies, future sequencing developments may also enable longer reads to be obtained without sacrificing quality. Longer reads at greater depth may help to string haplotypes together, increasing genomic coverage (Table 1). Long-range PCR has been used to help achieve this and other technologies, such as nanopore sequencing, might also facilitate long-read sequencing projects for eDNA [126,168].

Once environmental nuDNA has been captured, statistical programs can be developed specifically to analyze environmental genetic data. It may be possible to use rarefaction methods to estimate how well genetic diversity has been sampled in a location, as has been used in eDNA metabarcoding studies to estimate taxa obtained [169]. Additionally, eDNA may be able to borrow and modify techniques from pool-seq methods. Pool-seq can provide cost-effective, accurate allele frequency estimations when given large population sizes (>20 individuals) with sufficient (>20×) coverage [150,170,171]. Reliable estimates of F_ST_, based on an analysis-of-variance framework, have been made using this technique to infer population structure using nuclear markers [171]. However, pool-seq may require statistical modelling to obtain allele frequencies [171], and because many individuals are often needed, may not work as well for low-density endangered species [150]. This technique cannot identify individuals and has not yet been fully developed for detecting intraspecific genetic diversity from multi-species mixtures and does not necessarily do well with mixtures that are not in equimolar concentrations [145,150,170,172]. Despite multi-species gaps and potential eDNA stochasticity, pool-seq techniques may be a way forward for statistically quantifying allelic variation between populations in environmental samples. Once eDNA samples have been sequenced, regardless of method, alleles that little depth, such as nuclear eDNA, may be haplodized for analysis [173,174]. Originally used in the ancient DNA (aDNA) field, haplodizing randomly or statistically selects a single allele from each locus for analysis purposes [173,174]. This provides a way forward for eDNA to be used with already established population genetic statistics [173]. Potentially, statistical models can be built for total variation detection at a specific locus [114]. This has already been done for taxon biodiversity [114,175,176]. Additionally, estimates of relatedness between populations could be based on genetic similarities [42,68,69]. If nuDNA eDNA techniques are developed for eDNA samples, it would broaden the range of biological questions that eDNA could be used to answer.

These exciting technological advances paired with environmental genetics techniques offer promising new ways to address broad evolutionary genetics questions without direct sampling. Amazing progress is occurring all the time in the use of environmentally derived genetic material. For example, despite the much more rapid degradation of RNA in the environment, even environmental RNA (eRNA) samples are now being used to provide a snapshot of the biological diversity and processes present at the community level (Table 1) [177,178,179]. Additionally, eRNA has identified foraminiferal taxa not found with eDNA techniques [180]. It is feasible to compare these eDNA and eRNA snapshots between populations to understand differential expression of genes in macrofauna at different sites [179]. Studying community-level genetics may also be a promising research avenue [181]. Universal primers can already target haplotypes of multiple species [70,72]. Once these technologies are further refined, eDNA and eRNA may become tools for examining selection at the community level [181]. By examining targeted gene expression with eRNA, documenting community responses to common pressures, such as anthropogenic climate shifts, may be possible. Furthermore, combining eDNA with aDNA techniques may give researchers a lens into past communities [182]. With new technology and human ingenuity driving unprecedented datasets, we can answer questions about the patterns and processes of how populations are changing in new ways that may be less harmful.

## 5. Conclusions

The need for non-invasive sampling, especially for low-density and cryptic populations, will continue to drive technological development. Environmental DNA population genetic methods have the potential to fill this need if developed and tested rigorously. Limited conservation funding and a need to minimize stress to wild animals, coupled with increasing molecular expertise, declining taxonomic expertise, and declining sequencing cost make eDNA an attractive genetic sampling tool [197,198,199]. We envision the next step in eDNA for population genetics as describing allelic variation between populations, using previously sequenced traditional samples as a reference. Sampling with eDNA will likely never completely replace invasive sampling because organism capture allows for the collection of additional and important non-genetic data, such as size, age, maturity, isotopic measurements, enzymatic activity, and hormone activity [200,201,202,203]. Population genetic eDNA methods can be a useful complement to traditional, direct-sampling studies; eDNA is another tool in the population genetic toolbox. While there are challenges facing eDNA population genetic methods, the field can profit from—and build upon—the framework of extensive technological development already associated with the fields of aDNA and forensics. Technological advances have facilitated studies of disease, migration, and population dynamics of past organisms with aDNA techniques that were not feasible even 20 years ago [204,205,206]. Environmental DNA research faces similar technological challenges to aDNA associated with the recovery and sequencing of degraded, low copy number target DNA. Improving conservation management tools is of critical importance in light of global environmental change [207,208]. We predict that eDNA technology will have a key role to play in providing rapid and broad scale insights into the population genetics of imperiled and difficult-to-sample species around the globe.

## Figures and Tables

**Figure 1 genes-10-00192-f001:**
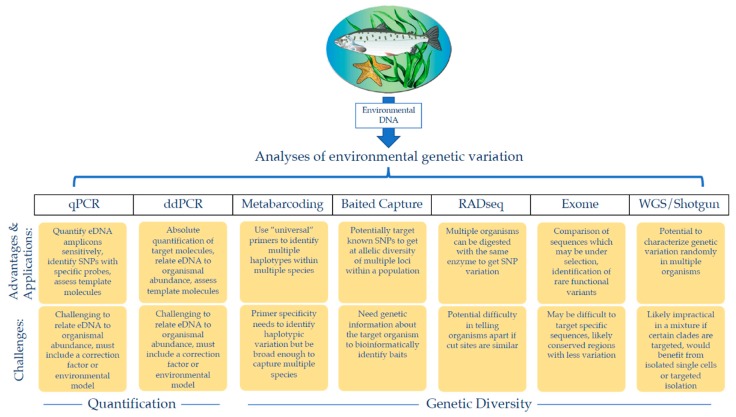
Possible avenues of analyzing environmental DNA (eDNA) for population information. After eDNA is shed into the water column by target organisms of interest, collected, and extracted, multiple analyses can be used to obtain population data. These analyses have diverse applications, highlighted in the top yellow row, but each technique also presents its own challenges, highlighted in the bottom yellow row.

**Table 1 genes-10-00192-t001:** Potential challenges facing environmental DNA population genetic research with suggestions for tools to help mitigate some of the challenges. While previous research has applied a number of these tools to these challenges (as cited), research still remains to be done to fully address how each tool can help mitigate each challenge, and to identify advantages and drawbacks of each. Note that not all tools and techniques apply to all challenges.

Challenges							
		Abundance	Allelic Drop-Out	Bioinformatic Challenges	Identifying Individuals	Long-Term Datasets	Obtaining Nuclear Markers
**Tools and Techniques**	**Automated sampling**	Reduce spatial and temporal variance, especially for difficult-to-sample areas. Standardized deployment may help detect abundance changes in regular intervals [183,184].			Document individual presence repeated through time [185].	Precise, standardized capture across time and space [186,187].	
	**Baited capture methodology**	Baited capture may reduce PCR amplification needed, reflecting true abundance ratios better. [188].	Targeted capture of specific allelic variation [189].	Identification of specific SNPs in a population.		Capture of specific allelic variation across time.	Target of nuDNA, especially SNP markers [189].
	**Droplet digital PCR**	Absolute quantification of target molecules [61,165,190].	Sensitively amplify different allelic variation, could reduce drop-out [191].	Provides absolute quantification of specific molecule abundance [192,193].	Perhaps amplify and quantify single-cell eDNA.	Sensitively quantify changes in target molecules over time.	Amplify and quantify nuclear marker loci.
	**eRNA**	Increase temporal resolution [177].	Increase temporal resolution of expressed alleles.	Identify allele-specific expression at the population level [194].	Detection of live individuals [195].	Examine expressed gene changes within a population or community [178,180].	Increased temporal resolution of nuclear genetic variation [194,195].
	**Increased sequencing depth**		Could increase detection chance of low copy number alleles [196]. Pool-seq may aid in elucidating allelic variation of large samples.	Could increase chance of detecting genetic diversity in replicates, perhaps allows for stricter filtering.	Could increase confidence in detection of individuals, especially if using single-cell techniques.	More robust datasets may show change throughout time at a finer scale.	Increased probability of detecting rare alleles.
	**Increased sequencing read length (nanopore, long-range mtDNA)**		Capture of long reads or mtDNA genomes, see which alleles are linked [126]	Longer reads may help compile individuals’ mtGenomes [126].	Links SNPs to form genomic or mtDNA haplotypes.	May see recombination patterns through time.	Increased genomic coverage [126].
	**Single-cell sequencing**	Approximation of unique individuals per sample assuming different genomes.	May have some allelic dropout if depth of sequencing is low [162].	Identification of individuals allows for information to be analyzed with traditional population-genetics methodology.	Identify individuals based on cell genome [156].	Identify changes in individual presence.	Target of nuDNA, perhaps even able to aid in sequencing of whole genome [155].
	**“Universal” primers specific enough for intraspecific variation**		Alleles of multiple species identified in the same sample with same primer [72].		Possibility to identify multiple individuals of multiple species if individuals can be sorted.	Multiple species targeted for community composition snapshots [70].	Possibility to target nuclear markers in multiple species in the same sample.

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
