# Peer review of "Beyond Biodiversity: Can Environmental DNA (eDNA) Cut It as a Population Genetics Tool?"

_genes, 2019, doi:10.3390/genes10030192_

Round 1

Reviewer 1 Report

In their Perspective article, Clare Adams and colleagues make the case that environmental DNA (eDNA) could be used for population genetics analyses.

The interest in the application of eDNA for ecological studies has shown a rapid expansion over the past decade. While eDNA has many advantages, there are also many drawbacks that have constrained the progression from presence/absence surveys, to quantitative population genetic studies. In particular, the problem that eDNA samples are essentially a pool of DNA from different individuals, with no easy way to tease out the relative contribution of each individual to the DNA within the sample. This negates (or complicates) the possibility to perform most individual-based population genetic statistics, for example those based on allele or haplotype frequencies. 

Adams et al. cover a lot of ground in this perspective and provide a comprehensive overview of the challenges of estimating population genetic metrics from eDNA. However, they are on the whole optimistic proponents of the potential for eDNA to be harnessed for population genetics.  My first impression when reading the title was that this paper would provide a road map to take the field that logical and significant step forward, and show that structure and other population genetics parameters could be detected and quantified using eDNA. But in it's present form I feel that the perspective has fallen a little short of this. Hence my recommendation of Major Revision. However, the article was very well written and would be a useful resource and I want to stress to the authors and editors that my impression of the manuscript was mostly positive and I was teetering between recommending major and minor revisions.

What comes across from the current draft is that the authors have a strong background in the laboratory methods to harness and generate sequence or genotype data from eDNA and provide a very high quality discussion of this aspect. However, I believe the major hurdle to moving beyond presence/absence or relative abundance estimates to population genetics applications using eDNA is the lack of a robust statistical framework to deal with the problem that samples consist of pools of multiple individuals (and potentially different species), which will not be present in equimolar quantities. I'd like to see the authors delve into the statistical problems associated with deriving population genetic inference from eDNA data. Perhaps, there would be lessons that could be borrowed from poolseq approaches often used in studies of organisms such as Drosophila, that are too small to economically individually genotype (see work by Christian Schlotterer's group). The ancient DNA field also encounters many of the same issues, with DNA extracted from a single ancient bone still essentially comprising of a metagenomic sample due to contaminating DNA, and suffering from low yields of target DNA and potential for allelic dropout and other quality associated artefacts. The ancient DNA field employs approaches such as 'haploidising' (randomly selecting a single allele from each site) or using approaches that provide a measure of uncertainty of the genotype at any given site, based upon the quality of the sequence data (.e.g. base and mapping quality) at that site, e.g. by estimating genotype likelihoods. I suspect the most realistic approach to incorporating eDNA data into population genetics studies will occur when there is already existing data from more traditionally collected samples to compare with (i.e. as per the Sigsgaard et al paper). Assignment of eDNA samples to populations, using population genetics approaches and a reference dataset of traditionally sampled and genotyped individuals seems to me to be more achievable than inferring population genetic parameters from eDNA samples alone. 

Overall, I believe that this a very worthwhile perspective, it is well-written and enjoyable to read.

I believe that if the authors could add some insights into the issues I've highlighted in the previous paragraph into the perspective, that it would round it out and make it suitable for publication. 

I have a few minor comments:

Line 46: There is also the extra effort of having to directly encounter the target species.

Line 89:  Rather than 'traditional tissue sample', I would suggest 'traditionally collected tissue samples' or  'directly sampled tissue'.

Line 96: Killer whales are marine mammals, so typically we refer to the water body not the land mass: Northeast Pacific killer whales.

Line 104:  'It may therefore be possible....'

Line 113-115, Explain in a bit more detail what these positive controls are and how they work, so that the reader does not have to look this up in the cited source papers.

Lines 130-156,Odd fit here, seem to be more akin to earlier text a few paragraphs prior.

Line 156, “will require some significant challenges to be overcome.” What are these challenges? See my main point.

Line 206 ‘changes’ should be ‘chances’

Line 223-224. Explain why nuDNA can show different patterns to mtDNA. You give one reason in the preceding sentence, and it may improve the flow of the paragraph if you switch the order of these two sentences.

Lines 226-230 require some referencing.

Figure 1. What is the delineation between Shotgun and WGS? Most WGS is achieved through Shotgun sequencing? or are you specifically referring to genome assembly from long-reads?

Author Response

In their Perspective article, Clare Adams and colleagues make the case that environmental DNA (eDNA) could be used for population genetics analyses.

The interest in the application of eDNA for ecological studies has shown a rapid expansion over the past decade. While eDNA has many advantages, there are also many drawbacks that have constrained the progression from presence/absence surveys, to quantitative population genetic studies. In particular, the problem that eDNA samples are essentially a pool of DNA from different individuals, with no easy way to tease out the relative contribution of each individual to the DNA within the sample. This negates (or complicates) the possibility to perform most individual-based population genetic statistics, for example those based on allele or haplotype frequencies. 

Adams et al. cover a lot of ground in this perspective and provide a comprehensive overview of the challenges of estimating population genetic metrics from eDNA. However, they are on the whole optimistic proponents of the potential for eDNA to be harnessed for population genetics.  My first impression when reading the title was that this paper would provide a road map to take the field that logical and significant step forward, and show that structure and other population genetics parameters could be detected and quantified using eDNA. But in it's present form I feel that the perspective has fallen a little short of this. Hence my recommendation of Major Revision. However, the article was very well written and would be a useful resource and I want to stress to the authors and editors that my impression of the manuscript was mostly positive and I was teetering between recommending major and minor revisions.

Thank you for your very helpful comments; we have revised the manuscript in line with the reviewer’s suggestions.

What comes across from the current draft is that the authors have a strong background in the laboratory methods to harness and generate sequence or genotype data from eDNA and provide a very high quality discussion of this aspect. However, I believe the major hurdle to moving beyond presence/absence or relative abundance estimates to population genetics applications using eDNA is the lack of a robust statistical framework to deal with the problem that samples consist of pools of multiple individuals (and potentially different species), which will not be present in equimolar quantities. I'd like to see the authors delve into the statistical problems associated with deriving population genetic inference from eDNA data. Perhaps, there would be lessons that could be borrowed from poolseq approaches often used in studies of organisms such as Drosophila, that are too small to economically individually genotype (see work by Christian Schlotterer's group).

Thank you for this insightful comment on adding in Pool-seq and directing us to a body of work by the Schlotterer lab. We have added in a paragraph that discusses pool-seq (Section 4, lines 322-338) and a paragraph on statistics and environmental samples not having eqi-molar concentrations. (Section 3, lines 246-248).

 The ancient DNA field also encounters many of the same issues, with DNA extracted from a single ancient bone still essentially comprising of a metagenomic sample due to contaminating DNA, and suffering from low yields of target DNA and potential for allelic dropout and other quality associated artefacts. The ancient DNA field employs approaches such as 'haploidising' (randomly selecting a single allele from each site) or using approaches that provide a measure of uncertainty of the genotype at any given site, based upon the quality of the sequence data (.e.g. base and mapping quality) at that site, e.g. by estimating genotype likelihoods.

We have added in a few sentences about haploidising (lines 338-343). This was a great comment.  

I suspect the most realistic approach to incorporating eDNA data into population genetics studies will occur when there is already existing data from more traditionally collected samples to compare with (i.e. as per the Sigsgaard et al paper). Assignment of eDNA samples to populations, using population genetics approaches and a reference dataset of traditionally sampled and genotyped individuals seems to me to be more achievable than inferring population genetic parameters from eDNA samples alone. 

We agree with your comment. Lead author, Clare Adams, is currently working on this for their PhD work. We have added in the comment, “We envision the next step in eDNA for population genetics as describing some allelic variation between populations, using previously sequenced traditional samples as a reference,” to lines 375-376.

Overall, I believe that this a very worthwhile perspective, it is well-written and enjoyable to read.

I believe that if the authors could add some insights into the issues I've highlighted in the previous paragraph into the perspective, that it would round it out and make it suitable for publication. 

Thank you for your support, we very much appreciate your comments.

I have a few minor comments:

Line 46: There is also the extra effort of having to directly encounter the target species.

We have changed the sentence on line 46 to say, “…and traditional sampling efforts to directly encounter the target species may be more costly than reagents required for eDNA analysis [22].”

Line 89:  Rather than 'traditional tissue sample', I would suggest 'traditionally collected tissue samples' or  'directly sampled tissue'.

We have changed the sentence on line 122 to, “… that were also known from directly sampled tissues.”

Line 96: Killer whales are marine mammals, so typically we refer to the water body not the land mass: Northeast Pacific killer whales.

We have changed the sentence on line 129 to, “More recently, seawater eDNA work on Northeast Pacific killer whales (Orcinus orca) ...”

Line 104:  'It may therefore be possible....'

Have changed the sentence in line 137.

Line 113-115, Explain in a bit more detail what these positive controls are and how they work, so that the reader does not have to look this up in the cited source papers.

To add clarification for the reader we have changed the sentences (lines 146-149).

Lines 130-156,Odd fit here, seem to be more akin to earlier text a few paragraphs prior.

We have moved the section, “Why use eDNA over direct sampling for population genetics?” to the end of the first section, now lines 86-115.

Line 156, “will require some significant challenges to be overcome.” What are these challenges? See my main point.

We have modified the sentence to read, “However, significant sequencing, bioinformatic, and statistical challenges will need to be overcome ...” (lines 109-111).

Line 206 ‘changes’ should be ‘chances’

Corrected, line 213.

Line 223-224. Explain why nuDNA can show different patterns to mtDNA. You give one reason in the preceding sentence, and it may improve the flow of the paragraph if you switch the order of these two sentences.

We have switched the order of these two sentences (lines 227-228).

Lines 226-230 require some referencing.

We have added in a paper on identifying individual wombats by their hairs (Sloane et al., 2000) and a paper on using microsatellites to identify the minimum number of individual leopards (Mondol et al., 2009), lines 235-238.

Figure 1. What is the delineation between Shotgun and WGS? Most WGS is achieved through Shotgun sequencing? or are you specifically referring to genome assembly from long-reads?

We have combined Shotgun and WGS in the same box in Figure 1. We had originally separated the two out because shotgun sequencing has previously yielded mostly bacterial genetics rather than metazoan genetics and single-cell techniques within a WGS approach may be more targeted than simply shotgun sequencing the whole eDNA sample.

Reviewer 2 Report

The manuscript, “Beyond biodiversity: can environmental DNA (eDNA) cut it as a population genetics tool?” by Adams et al. report on using eDNA to study population genetics, while suggesting methods for future studies.   

Overall, this manuscript is well written and is laid out well. The data presented are interesting and are a valuable contribution to the field of eDNA. In my opinion, this manuscript fits well within the scope of Genes.

I would ask the authors to also include information and literature in their introduction about the difficulty of sampling rare, either naturally rare or endangered, species around line 44. I think this helps to build the case for this method earlier even though there is discussion of this later. 

I would also like the authors to discuss and include the results from Yamamoto et al. 2017. “Environmental DNA metabarcoding reveals local fish communities in a species-rich coastal sea.” Yamamoto demonstrated that increased PCR replicates and subsequent sequencing of these replicates increased the number of species detected.

Line 326: how about also including the decline in taxonomic experts?

Please see specific comments below:

Line 156: Sentence structure is a bit awkward. How about changing the structure by starting with “significant challenges will need to be overcome”

Author Response

The manuscript, “Beyond biodiversity: can environmental DNA (eDNA) cut it as a population genetics tool?” by Adams et al. report on using eDNA to study population genetics, while suggesting methods for future studies.   

Overall, this manuscript is well written and is laid out well. The data presented are interesting and are a valuable contribution to the field of eDNA. In my opinion, this manuscript fits well within the scope of Genes.

I would ask the authors to also include information and literature in their introduction about the difficulty of sampling rare, either naturally rare or endangered, species around line 44. I think this helps to build the case for this method earlier even though there is discussion of this later. 

Thank you so much for your comments! We have included new sentences on lines 46-50.

I would also like the authors to discuss and include the results from Yamamoto et al. 2017. “Environmental DNA metabarcoding reveals local fish communities in a species-rich coastal sea.” Yamamoto demonstrated that increased PCR replicates and subsequent sequencing of these replicates increased the number of species detected.

We have now included a reference to this work in line 187.

Line 326: how about also including the decline in taxonomic experts?

We have modified this sentence to include this point (line 374).

Please see specific comments below:

Line 156: Sentence structure is a bit awkward. How about changing the structure by starting with “significant challenges will need to be overcome”

We have reworked the sentence, now lines 109-111. Thanks for the help!

Reviewer 3 Report

This review paper is well researched and highly topical. It provides an excellent overview of eDNA and discusses its potential use in population genetics. My search has not revealed another review on the topic and there has been sufficient advances to warrant such a review. It is also an important way to encourage conservation scientists to consider the use of eDNA in population genetic studies. The authors also do a very good job at acknowledging and discussing the current difficulties that this approach is presented. One possible improvement is to have a box explaining certain promising advances such as strict sequencing read filtering in more detail.

Author Response

This review paper is well researched and highly topical. It provides an excellent overview of eDNA and discusses its potential use in population genetics. My search has not revealed another review on the topic and there has been sufficient advances to warrant such a review. It is also an important way to encourage conservation scientists to consider the use of eDNA in population genetic studies. The authors also do a very good job at acknowledging and discussing the current difficulties that this approach is presented. One possible improvement is to have a box explaining certain promising advances such as strict sequencing read filtering in more detail.

Thank you for the encouraging comments. While no standardized filtering exists yet for environmental DNA population genetics, and not even really for eDNA in general as it is highly dataset dependent, we try to push the reader toward using positive controls to inform what sequence read filtering will work for them (eg. lines 195-198). We have added the sentence, “This will aid in determining appropriate filter parameters to ensure strict filtering for high quality data without being too restrictive and ignoring real biological signal.” On lines 304-305.

Reviewer 4 Report

Dear Editor, dear Authors,

The authors have identified a topic of central importance in environmental research: How can we pull out more data from the treasure that eDNA provides. Thus, a timely manuscript with a very thorough literature analysis. Definitely, there is a place for such a study in a prominent journal. In general I would more recommend a journal such as "Molecular Ecology", "Environmental DNA", "Metabarcoding and Metagenomics” or “Heredity" for such an article, as its probably mostly environmental researchers interested in this topic of eDNA. But this is my personal opinion and I assume the authors have carefully thought about the journal choice. I think the authors have done a very good job mostly, but I have two central points to raise and several minor points:

1)  The strongest argument for single specimen based population genetics is the possibility to get individual-based genotypes (homo-/heterozygosity at individual loci). With all the eDNA based approaches the data will more be of a poolseq-approach, i.e. studying allele frequencies. However, population genetic theory is mostly founded on linking allelic and genotypic data (obs vs. expected). These are the basis for many . There are the basis many/most inferences such as in BAPS, STRUCTURE, r2 calculations etc. - testing expectations of heterozygosity distributions with allele variants. It should be stated more clearly that this can very unlikely be obtained with eDNA. The arguments are very similar to the poolseq discussions and I think the manuscript should more strongly seek the similarities of all approaches with poolseq (which is also without doubt a very strong technique but mostly limited to allele frequency changes).

2) I feel that both the figure and table and not well-connected to the text. Why e.g. WGS and Shotgun in figure different not clear to me. You mention exome sequencing in the text, but not shown here. What does qPCR precisely deliver should be elaborated a bit further. Why only ampliconts and ddPCR absolute quantificaiton of target molecules. To me, both approaches try to assess template moleculs. Also check the “wordish” red underlyings in the text of the figure. Also I do not fully agree with the gradient (more commonly used - emergent). I think ddPCR is not as prominent as WGS. Bait capture seem to me very ’new’ in the field (at least environmental analysis). But as this is a pespective paper we may disagree here I assume. Important is to mention the different field a bit more thoroughly. Also several aspects in the table are rather little explained. E.g. eRNA gene expression changes - further remarks on the precise challenge (lack of references to link the gene expression data). Several other field might be populated, e.g. baited caputre methodology and abundance- there is a study that at least suggests that baits data can do better here (what I do not fully read from the data published here: https://www.biorxiv.org/content/biorxiv/early/2016/11/13/087437.full.pdf) So while I have no definite answer here I want you to carefully think how table 1 and figure 1 can be improved in the global context.

Minor comments:

- Add some more information about numts. Rare variants and numts can be mixed up when jumping on mitochondrial DNA as a proxy for population genetics.

- L163 check formatting.

- L170: I think DADA2 is more “upstream” in the process. You would first use DADA2 and then BAPS, adegenet etc. Please delete, you may add other software though here that does downstream analysis.

- L194: Please check if you cited ref 102 correctly; I think it actually suggested the opposite. A general reference to the topic would be Elbrecht and Leese (2015) https://doi.org/10.1371/journal.pone.0130324

- L202: e.g. 

- L215: space after >16

- L222: e.g. (not eg.) but also I doubt ref 82 is an ideal reference to what you want to cite. More appropriate in the context of mito-nuclear discordances e.g. Weigand et al. (2017) https://doi.org/10.1111/mec.14292

- L237: RADseq here or RADSeq in the figure.

Author Response

Dear Editor, dear Authors,

The authors have identified a topic of central importance in environmental research: How can we pull out more data from the treasure that eDNA provides. Thus, a timely manuscript with a very thorough literature analysis. Definitely, there is a place for such a study in a prominent journal. In general I would more recommend a journal such as "Molecular Ecology", "Environmental DNA", "Metabarcoding and Metagenomics” or “Heredity" for such an article, as its probably mostly environmental researchers interested in this topic of eDNA. But this is my personal opinion and I assume the authors have carefully thought about the journal choice. I think the authors have done a very good job mostly, but I have two central points to raise and several minor points:

Thank you. We had thought about these journals but ultimately decided to go with GENES because we were invited to write something for the special issue on Conservation Genomics and Genetics. We believe that eDNA could potentially help with conservation monitoring.

1)  The strongest argument for single specimen based population genetics is the possibility to get individual-based genotypes (homo-/heterozygosity at individual loci). With all the eDNA based approaches the data will more be of a poolseq-approach, i.e. studying allele frequencies. However, population genetic theory is mostly founded on linking allelic and genotypic data (obs vs. expected). These are the basis many/most inferences such as in BAPS, STRUCTURE, r2 calculations etc. - testing expectations of heterozygosity distributions with allele variants. It should be stated more clearly that this can very unlikely be obtained with eDNA. The arguments are very similar to the poolseq discussions and I think the manuscript should more strongly seek the similarities of all approaches with poolseq (which is also without doubt a very strong technique but mostly limited to allele frequency changes).

We have now included a paragraph about statistical methods (lines 241-255). We have also included a sentence about the inability to obtain individual heterozyosity with eDNA, unless single-cell techniques are used (lines 253-254). While we do believe that the next step in eDNA pop gen will likely be more akin to Pool-seq, outlined in section 4, lines 333-340, and we also contend that single-cell sequencing and rapidly advancing cell-sorting technologies may eventually be used to sequence individuals (lines 283-290).  

We include the sentence, “It is unlikely that individual heterozygosity will be able to be obtained with eDNA, as eDNA often contains fragmented DNA which cannot be assigned to a specific cell” on lines 251-253.

2) I feel that both the figure and table and not well-connected to the text. Why e.g. WGS and Shotgun in figure different not clear to me. You mention exome sequencing in the text, but not shown here. What does qPCR precisely deliver should be elaborated a bit further. Why only ampliconts and ddPCR absolute quantificaiton of target molecules. To me, both approaches try to assess template moleculs. Also check the “wordish” red underlyings in the text of the figure. Also I do not fully agree with the gradient (more commonly used - emergent). I think ddPCR is not as prominent as WGS. Bait capture seem to me very ’new’ in the field (at least environmental analysis). But as this is a pespective paper we may disagree here I assume. Important is to mention the different field a bit more thoroughly.

We have combined WGS and Shotgun sequencing in the figure. We have added in exome sequencing. While we were not 100% sure what was being referred to in terms of qPCR precisely delivering, we added in Library amplification as qPCR is used in our lab for this purpose. We have added in “assess template molecules” into qPCR and ddPCR’s Advantages and Applications. (Figure 1)

We have changed the gradient to partitions of Quantification and Genetic Diversity. Some of the authors on this paper (Knapp and Bunce) come from an aDNA background, so baits weren’t as new to us as Pool-seq, but we suspect that is simply a matter of perspective. We hope that by changing the gradient (commonly used -> emergent) to Quantification and Genetic Diversity will be more satisfactory. (See figure 1).

Also several aspects in the table are rather little explained. E.g. eRNA gene expression changes - further remarks on the precise challenge (lack of references to link the gene expression data). Several other field might be populated, e.g. baited caputre methodology and abundance- there is a study that at least suggests that baits data can do better here (what I do not fully read from the data published here: https://www.biorxiv.org/content/biorxiv/early/2016/11/13/087437.full.pdf) So while I have no definite answer here I want you to carefully think how table 1 and figure 1 can be improved in the global context.

We’ve added in the baited capture and abundance with a reference (not from bioRxiv, but from Liu et al 2016; Mitochondrial capture enriches mitoDNA 100 fold, enabling PCRfree mitogenomics biodiversity analysis; https://doi.org/10.1111/1755-0998.12472. We’re not entirely clear on what is meant regarding the eRNA gene expression changes, but we’ve added in more references to the table, especially for eRNA. We have also referred to Table 1 in the text at lines 282, 285, 300, 307, 317, and 319.

Minor comments:

 - Add some more information about numts. Rare variants and numts can be mixed up when jumping on mitochondrial DNA as a proxy for population genetics.

We have added in a few lines (lines 231-232) about numts.

- L163 check formatting.

Correct and extra citation removed.

- L170: I think DADA2 is more “upstream” in the process. You would first use DADA2 and then BAPS, adegenet etc. Please delete, you may add other software though here that does downstream analysis.

We have removed DADA2 (line 177).

- L194: Please check if you cited ref 102 correctly; I think it actually suggested the opposite. A general reference to the topic would be Elbrecht and Leese (2015) https://doi.org/10.1371/journal.pone.0130324

We have changed this to cite Elbrect and Leese 2015, lines 192.

- L202: e.g. 

 Corrected (line 288).

- L215: space after >16

Corrected, line 224.

- L222: e.g. (not eg.) but also I doubt ref 82 is an ideal reference to what you want to cite. More appropriate in the context of mito-nuclear discordances e.g. Weigand et al. (2017) https://doi.org/10.1111/mec.14292

 Corrected, line 231.

- L237: RADseq here or RADSeq in the figure.

Edited to be RADseq, line 264.

Round 2

Reviewer 1 Report

In this revised submission the authors have thoughtfully and thoroughly addressed all my comments based upon the previous draft. I now find the manuscript suitable for publication in Genes and feel confident that this perspective will be a useful reference tool for those working in the field of eDNA.

Congratulations to the authors. 

Reviewer 2 Report

The authors have made the suggested changes and I believe this manuscript should be accepted for publication.